# A Gait-Based Real-Time Gender Classification System Using Whole Body Joints

**DOI:** 10.3390/s22239113

**Published:** 2022-11-24

**Authors:** Muhammad Azhar, Sehat Ullah, Khalil Ullah, Ikram Syed, Jaehyuk Choi

**Affiliations:** 1Department of Computer Science & IT, University of Malakand, Chakdara 18800, Pakistan; 2Department of Software Engineering, University of Malakand, Chakdara 18800, Pakistan; 3School of Computing, Gachon University, 1342 Seongnam-daero, Sujeong-gu, Seongnam-si 13120, Republic of Korea

**Keywords:** gender identification, human gait, machine learning, feature selection

## Abstract

Gait-based gender classification is a challenging task since people may walk in different directions with varying speed, gait style, and occluded joints. The majority of research studies in the literature focused on gender-specific joints, while there is less attention on the comparison of all of a body’s joints. To consider all of the joints, it is essential to determine a person’s gender based on their gait using a Kinect sensor. This paper proposes a logistic-regression-based machine learning model using whole body joints for gender classification. The proposed method consists of different phases including gait feature extraction based on three dimensional (3D) positions, feature selection, and classification of human gender. The Kinect sensor is used to extract 3D features of different joints. Different statistical tools such as Cronbach’s alpha, correlation, *t*-test, and ANOVA techniques are exploited to select significant joints. The Coronbach’s alpha technique yields an average result of 99.74%, which indicates the reliability of joints. Similarly, the correlation results indicate that there is significant difference between male and female joints during gait. *t*-test and ANOVA approaches demonstrate that all twenty joints are statistically significant for gender classification, because the *p*-value for each joint is zero and less than 1%. Finally, classification is performed based on the selected features using binary logistic regression model. A total of hundred (100) volunteers participated in the experiments in real scenario. The suggested method successfully classifies gender based on 3D features recorded in real-time using machine learning classifier with an accuracy of 98.0% using all body joints. The proposed method outperformed the existing systems which mostly rely on digital images.

## 1. Introduction

The term “gait” describes a person’s manner of walking. Every person has a unique gait that can be used as a behavioral trait to identify them. Since gait capture may be performed from a distance, the system does not require any assistance from the person. Gait recognition has therefore become more common in recent years and has a variety of applications, including forensics, security systems, surveillance, and the investigation of criminal activity. As CCTV is currently implemented in nearly all public areas [1,2,3], it unobtrusively records the person’s gait.

In the existing research work, the majority of the literature has focused on gait recognition [4,5,6,7,8,9,10,11,12,13,14], whereas gender classification based on gait offers a lot of potential for real-time applications [15]. The difficulty of discovering lost people, especially youngsters, at train stations, airports, bus stops, and other public locations can be solved with the help of gait-based gender classification, which can act as a clue in an investigation. The gender classification based on gait can also be applied in commercial activities. For instance, relevant advertisements can be shown on the digital screen if the gender of the person entering the store is known. A talking robot that has been installed in supermarkets can recognize customers through cameras and assist them with gender-specific needs including finding washrooms, food stores, men’s wear stores, and women’s wear stores.

The majority of gender classification studies make use of a different number of biometric features for gender classification, e.g., Pries et al. [16] used 13 features while Alharbi et al. [17] used 7 different joints. In the literature, few joints are often used to achieve gait-based gender recognition; therefore, little focus has been given to studying the gender differences based on all joints while walking. Unfortunately, very few studies in the recent research literature have used a self-created dataset for gender classification, which may be thought of as the main motivation for this work.

### 1.1. Research Gap and Motivation

The existing gait recognition approaches have compared limited joints of male and female during gait and less focus has been paid for complete study/analysis of joints.Gender has been classified based on gait in recent research [18,19,20,21,22], although the accuracy of this classification mostly depends on 2D camera images.Many researchers have recognized the gender of walking participants [23,24,25,26], but they have analyzed the results using a comparatively limited dataset, which may adversely affect the system’s accuracy.

### 1.2. Main Contributions

The following are the key innovations and contributions of our work:Using information obtained from MS Kinect, the authors created their own data set for gender classification.Applying a straightforward binary logistic regression model to the challenging issue of gender classification with better accuracy.The suggested system provides better accuracy as compared to the state-of-the-art system using the same data set.

In short, there is little information on gender-related differences in joints during casual walk; therefore, the purpose of this study is to compare/investigate every joint difference that contributes during walking. Finally, the gender of the walking is classified based on the complete set of joints positions. This paper is organized as follows. Section 2, presents related work, Section 3 describes proposed work, and Section 4 describes experiments and results. Finally, Section 5 is about conclusion and future work.

## 2. Related Work

Gender classification is performed in a number of ways using gait. According to the format of the gait, the existing gait identification techniques can be classified into 2D-based and 3D-based techniques [27,28,29]. The 2D-based gait identification techniques depend on a human silhouette, and is often collected by a single RGB camera in video surveillance. RGB-image-based methods are mostly used in this area of gait recognition. These methods are often categorized as model-free and model-based [18,23,30,31,32].

Model-free methods are often referred to as appearance-based techniques. The silhouettes, which are removed from the video sequences via background subtraction, are used to directly generate gait signatures. A gait energy image (GEI) is the most common representation of gait based on appearance [6], whereas 3D-based approaches typically make use of numerous calibrated cameras or cameras with depth sensors to extract 3D gait data. Moreover, 2D-based methods are typically simpler to execute because they simply require a normal camera. The 3D skeleton model depicted was suggested by Zhao et al. [33] and is based on 10 joints and 24 degrees of freedom (DOFs) recorded by various cameras. Yamauchi et al. [34] captured the dense 3D range gait, which may be used to identify people in various orientations. A system with four calibrated and synchronized cameras was developed by Krzeszowski et al. [35] to predict 3D gait motion from video sequences and identify view-variant gaits.

In terms of identifying gaits, both 2D-based and 3D-based approaches offer significant advantages; however, 3D-based approaches are more realistic as compared to 2D-based/image-based approaches because their robustness against the variation of observation perspective [36], walking speed [37], and clothing [38] is better than a 2D-based system.

Recently, the majority of researchers have focused on the view-invariant Kinect-based gait detection method because it is challenging to extract viewpoint information from only silhouettes [39]. In order to identify persons, Kale et al. [40] introduced a gait recognition algorithm based on static body metrics that are collected from the walking across multiple views. A framework was presented by Jean et al. [41] to compute and assess the view-normalized trajectories of the head and feet as seen in monocular videos. Hu et al. [42] suggested using a unitary linear projection to reduce the dimensions of the original gait parameters that were taken from any view(s) and increase their capacity to differentiate between different views. Because of its ability to collect skeletons with less distortion from clothes and lighting than prior depth sensors, the Kinect has become a popular tool for gait detection. Sivapalan et al. [43] used the Kinect sensor to convert the GEI from a 2D to a 3D representation. The lengths of body parts were calculated from joint positions and used as static anthropometric features by Araujo et al. [44] for gait recognition. The coordinates of all the joints recorded by the Kinect were used to produce an RGB image, which was then combined into a video to represent the walking sequence. This method was used by Milos et al. [45] to detect the gait. Price et al. [16] evaluated 11 skeletal features, such as step length and speed, which were used by Kinect as dynamic features; both static and dynamic features were combined for recognition. Yang et al. [46] developed relative distance-based gait features as an alternative to anthropometric measurements, which are better able to retain the periodic patterns of gait. Ahmed et al. [47] developed a gait signature using the total number of joint relative angles (JRA) sequences throughout a walking cycle. In order to determine the distance between two JRA sequences, they also created a dynamic temporal warping (DTW)-based kernel. Kastaniotis et al. [48] proposed a strategy for leveraging the Kinect for gait-based recognition. A person can be identified with 90% accuracy using a Kinect-based gait recognition system, as suggested by [16], which uses 13 biometric parameters. Similarly, X et al. [49] used 18 dynamic features that covered the dynamic angles in the lower limp joints. The system’s accuracy was still just 43.6%. A similar approach [50] used 14 features to identify objects, mostly based on the horizontal distance feature (HDF) and vertical distance feature (VDF). The system’s recognition accuracy was 92% when using the KNN method. An SVM-based method for gender classification using the Kinect obtained an average accuracy of 87% [51]. Azhar et al. used a limited set of joints in [52] for gender classification in real-time using Kinect sensor. With five subjects, another human identification system that used skeleton data from the Kinect sensor showed an accuracy of 86% [53]. Similarly, the researchers in [47] used joint relative angles (JRA) technique for 3D skeleton Kinect-based gait identification system with a 93.3% recognition rate. A different gait recognition and gender classification method based on the Kinect was suggested in [54]. The KNN classifier produced results with a reasonable accuracy of 96%. A gait-based recognition system for identifying individuals in baggy and lengthy clothing was proposed in [17]. Using nine joints the system used SVM and claimed a 97% accurate result.

The existing gait recognition approaches have compared limited joints of male and female during gait and less focus has been paid for complete study/analysis of joints. Furthermore, the majority of researchers have created systems that classify both age and gender in a single system, but they have typically divided age into various groups (classes), creating the issue of a classification system with a restricted age range. The key innovations and contributions of our work is the creation of our own dataset, comparison of all body joints of male and female using different statistical tools, comparison of the existing systems with the state-of-the-art systems and applying a straight forward binary logistic model to the challenging issue of gender classification with better accuracy.

## 3. Proposed System

This paper presents a comparative method for calculating the contribution of male and female body joints in walking. The contribution of male body joints is then compared to the contribution of female body joints using statistical methods such as correlation, *t*-test and ANOVA techniques. Then, Cronbach’s alpha technique is applied to find the reliability of joints. The findings also include a comprehensive list of all human body joints and their roles in gender classification. Furthermore, the proposed system model has two working stages i.e., training and testing. In the training phase, the 3D position of the gender are recorded through MS Kinect to find the best fitted model for gender classification. Finally, in the testing phase, the gender of the walking subject is recorded using percentage of the most contributed body joints via the binary logistic-regression-based machine learning model. The proposed system model is shown in Figure 1.

### 3.1. Dataset Generation

Since no other datasets provided both skeleton joint position and sufficient gender information, we had to develop our own. In this study, 373 participants ranging in age from 7 to 70 took part in the study. Inside the Kinect recognition space, the authors instructed each participant to walk in a free position while facing various directions (see Figure 2).

A cheap MS Kinect V1 camera is utilized to record the three dimensional (3D) joint movements of subjects. This sensor not only captures the traditional RGB image of the object being shown, but also the depth information between the objects and the camera. The Microsoft Kinect sensor is made up of a 3D depth sensor, RGB camera, tilt motor, and a Microsoft array. The Kinect sensor extracted the human skeleton 3D joints using the Kinect software development kit (SDK). The Kinect SDK uses Kinect skeletal tracking algorithm for extracting the 20 human joints. As seen in Figure 3, MS Kinect offers a full set of 20 usable joints, ranging from the head to the foot. Multiple records for each individual were taken in order to evaluate MS Kinect’s accuracy. The measurement was calculated from several distances, and 6 feet was found to be the ideal distance. The MS Kinect had a height of 3.4 feet and an angle of 0 degrees at 30 frames per second (fps). Every volunteer was free to adopt any behavior or stance at any time. There was no time restriction because the feature extraction process used the average of all frames.

### 3.2. Pre-Processing of Data

Our system for recording human gait consists of a single Kinect depth camera installed on a tripod and placed in a naturally lighted room. Humans are seen moving smoothly along the length of the room as the camera records their frontal motion and walking direction. Because the field of vision is 3 m and the depth field is 4.5 m, longer sequences can be captured. The Kinect makes it possible to capture a person at 30 fps and extract their 20-joint skeleton in real time; however, as our focus is on developing a reliable and effective gait processing approach, simple and generic low-cost acquisition setups are sufficient. A motion sequence in time t[1…T] is represented by a collection of skeletons, each of which has 20 joints. We use technique to represent the jth joint at time *t*.
(1)$$\begin{array}{c} P_{i}^{t} \end{array}=\left[\begin{array}{c} x_{j}^{t}\\ y_{j}^{t}\\ z_{j}^{t} \end{array}\right]$$

Equation (Equation 1) indicates that data can be recorded in 3D i.e., *x*, *y*, and *z* axis. We use the observation that the Euclidean distance between two joints should remain constant while the subject moves to create a scale-invariant skeleton. As a result, we normalize the skeleton joints in relation to how far they are from each other. Euclidean distance is a logical approach to compare two joints. The Euclidean distance between two joints of a body of a person during walk is defined as
(2)$$d_{rs}^{2}={∥{\underline{x}}_{r}-{\underline{x}}_{s}∥}^{2}$$

#### 3.2.1. Variance Covariance Matrix

In order to describe our gait data, we encode body joint data using a covariance matrix which calculates variance and covariance between various joints. We describe joints using a local relative coordinate system to achieve efficiency and consistency of body positions. In order to measure efficiency of each joint the sample variance–covariance matrix or covariance matrix is used as
(3)$$\begin{array}{c} S \end{array}=\left[\begin{array}{ccc} s_{11} & s_{12}\cdots & s_{1p}\\ s_{21} & s_{22}\cdots & s_{2p}\\ s_{p1} & s_{p2}\cdots & s_{pp} \end{array}\right]$$

The main diagonal values are referred to as the variances of particular joints and the remaining joints are the co-variances. The sample covariance matrix is calculated using the followingformula.
(4)$$S=\frac{1}{n}X^{\prime }.X-\bar{\underline{X}}.{\bar{\underline{X}}}^{\prime }$$
where *S* is covariance matrix, $\bar{\underline{X}}$ is the mean vector and *X* is a data matrix of joints, and where
(5)$$\begin{array}{c} \bar{\underline{X}} \end{array}=\left[\begin{array}{c} \bar{x_{1}}\\ \bar{x_{2}}\\ \vdots \\ \bar{x_{p}} \end{array}\right]$$

#### 3.2.2. Correlation Matrix

The correlation coefficient between ith and jth joints is given by
(6)$$r_{ij}=\frac{s_{ij}}{\sqrt{s_{ii}s_{jj}}}$$

The p×p matrix of correlation for all joints is defined as
(7)$$\begin{array}{c} R \end{array}=\left[\begin{array}{ccc} r_{11} & r_{12}\cdots & r_{1p}\\ r_{21} & r_{22}\cdots & r_{2p}\\ r_{p1} & r_{p2}\cdots & r_{pp} \end{array}\right]$$
where $R^{\prime }=R,r_{ij}=r_{ji}$ for i≠j and $-1\leq r_{ij}\leq +1$. If R=I, then all joints are uncorrelated.

Alternately, some of the fundamental statistical techniques are used to assess the contribution and effectiveness of joints. In order to identify all skeletons as having the same component length and compute an average value for each of its skeletal parts, we also compute the joints’ mean, variance, and coefficient of variation in *x*, *y*, and *z* coordinates, respectively.
(8)$${\bar{X}}_{Hip_{L}}=\frac{1}{n}\sum_{i=1}^{n}\left(X_{i}\right)$$
where i=1,2,3…n and ${\bar{X}}_{Hip_{L}}$ represents arithmetic mean of hip-left joint. The arithmetic mean for other axis i.e., *y* and *z* axis can be determined in the same way using Equation (Equation 4). Further, to compute a single value of various joints i.e., combined mean of *x*, *y*, and *z* coordinates of individual person for comparison of male and female joints during gait. The combined mean is statistically defined for joint Hip left as follows:(9)$${\bar{X}}_{C}=\frac{1}{3}\left({\bar{X}}_{Hip_{L}}+{\bar{Y}}_{Hip_{L}}+{\bar{Z}}_{Hip_{L}}\right)$$
where ${\bar{X}}_{C}$ is the combined mean of Hip left joint and ${\bar{X}}_{Hip_{L}}$, ${\bar{Y}}_{Hip_{L}}$ and ${\bar{Z}}_{Hip_{L}}$ are the means for joint Hip left of *x*, *y* and *z* coordinates.

#### 3.2.3. Variance of and Pooled Variance of Joints

To capture variation of joint features from person to person of male and female joints during gait, the authors calculated variances and standard deviation. In this phase, variance of all joints are calculated in the 3D axis e.g., variance of the left hip joint in *x*-axis is measured using Equation (Equation 10).
(10)$$\sigma_{XHip_{L}}^{2}=\frac{1}{N}\sum_{i=1}^{N}{\left(X_{i}-\mu_{X}\right)}^{2}$$
where *N* is the size or number of observation in the population, $\sigma^{2}$ is the variance of population, $X_{i}$ is the variables, and $\mu_{X}$ is the population mean at *x* coordinate. Similarly, the variance of hip left joint in *y* and *z* coordinates can be computed in the same way. The authors calculated the variances of other joints using Equation (Equation 11). After computing the variance of each joint, a pooled variance of all joints is calculated as
(11)$$\delta_{P}^{2}=\frac{1}{N}\left(N_{1}\delta_{1}^{2}+N_{2}\delta_{2}^{2}+N_{3}\delta_{3}^{2}+\ldots +N_{k}\delta_{k}^{2}\right)$$
where $\delta_{P}^{2}$ is the pooled variance while $\delta_{1}^{2}$, $\delta_{2}^{2}$, $\delta_{3}^{2}$*…*$\delta_{k}^{2}$, are the separate variances of joints. Similarly $N=N_{1}$ + $N_{2}$ + $N_{3}$, where *N* represents size of all joints and $N_{1}$, $N_{2}$, and $N_{3}$ are the sizes of *x*, *y*, and *z* axis. The pooled variances for joints $Hip_{L}$ may be computed as
(12)$$\delta_{PHip_{L}}^{2}=\frac{1}{N}\left(N_{1}\delta_{XHip_{L}}^{2}+N_{2}\delta_{YHip_{L}}^{2}+N_{3}\delta_{ZHip_{L}}^{2}\right)$$
where $N_{1}$, $N_{2}$, and $N_{3}$ are the sizes of *x*, *y*, and *z* axis respectively. $\delta_{PHip_{L}}^{2}$ is the pooled variances of joints $Hip_{L}$. Similarly, pooled variances of other joints can also be computed in the same way.

#### 3.2.4. Coefficient of Variation of Joints

The coefficient of variation (C.V) is used to assess the consistent joints out of the twenty joints for both male and female. The coefficient of variation is defined as
(13)$$C.V\left(X_{Hip_{L}}\right)=\frac{S.D(X_{Hip_{L}})}{mean(X_{Hip_{L}})}*100$$
where S.D represents the standard deviation of the hip left joint and the mean corresponds to the mean of the hip left joint.

#### 3.2.5. Average Variation in Particular Joints of Human during Gait

Mean, standard deviation, and coefficients of variation of male and female joints were computed by using various statistical software’s.

The results show comparatively a minimum score (10.31 and 9.05) for the coefficient of variation for the head joint of males and females. It suggests that the joint head is consistent joint for both male and female during the walk but more consistent for females as compared to males. On the other hand, C.V for the joint foot left of a male is a maximum of 17.58 as compared to the other joints, while C.V for the joint foot right of a female is also a maximum of 17.82 compared to other joints. It proves that the joints foot left and foot right of a male and a female vary more from individual to individual. C.V for the joints hand left and knee right of a male are same and also the C.V for joints spine and hip center of a female are the same, which show that these joints play the same role during gait. Table 1 shows the mean, standard deviation (S.D), and coefficient of variation (C.V) for various joints of male and female.

### 3.3. Comparison of Male and Female Joints

To assess the mean difference among all joints of male and female during the walk, different approaches are used.

#### 3.3.1. Comparison of Male and Female Joints by Correlation

To identify the relationship between various joints and various directions during walk, the author used the Pearson correlation method. Pearson correlation is a statistical method used to measure the relationship between any two variables. The correlation is always lies between −1 and +1, the correlation near to +1 indicates that there is strong and direct relationship between variables and correlation near to −1 indicates that there is strong and indirect relationship. When the correlation is near to zero indicates that there is weak relationship between variables. The Pearson correlation is defined as
(14)$$r=\frac{\left(n\sum xy-\sum x\sum y\right)}{\sqrt{\left[\left(n\sum x^{2}-{\left(\sum x\right)}^{2}\right)\left(n\sum y^{2}-{\left(\sum y\right)}^{2}\right)\right]}}$$

To know the difference between male and female joints during walk, the authors check the correlation between various joints. The significant difference between male and female joints was obtained from the correlation and there is high difference in the male and female joints during walk.

According to Table 2, there are strong and weak relationships between different joints of males and females during gait. During a walk, the relationship between *x* and *y* directions at HIP-center is 84% for males, while for females is 14%. Males in the HIP-center have a relationship between *y* and *z* directions of 88%, while females have 20%. Males at HIP-Center have 90% relationship between *x* and *z* directions, while females have 80%. The relationship between different joints in men and women is different. The relationship between *x* and *z* directions at joint Wrist left for males is 94%, while for females it is 79%. The relationship between *x* and *z* directions at joint Hand left for males is 93%, while for females is 72%. Similarly, the relationship between *x* and *z* directions at joint Knee right for males is 91%, while for females is 79%. The relationship between *x* and *z* directions at joint Ankle right is 93% for males, while 82% for females. The relationship between *x* and *z* directions at joint foot right is 93% for male joints, while for female joints, it is 88%. At joint HIP-Left, the correlation for males is 94% while for females it is 88%, which is a strong and significant relationship. The correlation results in Table 2 indicate that there is a significant difference between male and female joints during gait.

#### 3.3.2. Comparison of Male and Female Joints Using *t*-test

To compare various joints of male and female during gait, the *t*-test is applied to identify the significant difference, i.e., at joint hip center there is significant difference between male and female walking. As each human can walk in three directions such as *x*, *y*, and *z*, therefore, we calculate mean of *x*, *y*, and *z* directions, respectively, and compute their combine mean. The combined means were compared by using *t*-test. Consider *t*-test of equal variances and not equal variances provide the same conclusion. The results of two samples *t*-test are shown in Table 3. The formulation of our hypothesis is given below.

**Hypothesis 1.** *There is no difference at joint hip left of males and females during walk, i.e.,*(15)$$\mu_{Hip_{L}\left(M\right)}=\mu_{Hip_{L}\left(F\right)}$$**Hypothesis 2.** *There is significant difference at joint hip left for males and females during walk, i.e.,*(16)$$\mu_{Hip_{L}\left(M\right)}\neq \mu_{Hip_{L}\left(F\right)}$$
where $\mu_{Hip_{L}\left(M\right)}$ is the mean of hip left for males and $\mu_{Hip_{L}\left(F\right)}$ is the mean of joint hip left for females. To compare each joint of males and females pair wise during gait, the *t*-test and ANOVA technique are used for all comparisons of all joints at the same time. Assume that variances are the same for both male and female joint contribution for *t*-test. The *t*-test is defined for analysis as
(17)$$T=\frac{\left({\bar{X}}_{1i}-{\bar{X}}_{2i}\right)}{s_{p}\sqrt{\frac{1}{n_{1}}}+\frac{1}{n_{2}}}$$
where ${\bar{X}}_{1}$ is the combined mean of each joint of male, ${\bar{X}}_{2}$ is combined mean of each joint for female. $n_{1}$ is the size of males and $n_{2}$ is the size of females. The degree of freedom for *t*-test is
(18)$$\nu =n_{1}+n_{2}-2$$
where ν represents the degree of freedom. $S_{p}$ is the pooled standard deviation for both male and female joints and is defined as
(19)$$S_{p}=\sqrt{\frac{\left(n_{1}-1\right)s_{1}^{2}+\left(n_{2}-1\right)s_{2}^{2}}{n_{1}+n_{2}-2}}$$
(20)$$S_{1}=\sum \frac{{\left(x_{i_{\left(M\right)}}-{\bar{x}}_{\left(M\right)}\right)}^{2}}{n_{1}-1}$$
(21)$$S_{2}=\sum \frac{{\left(x_{i_{\left(F\right)}}-{\bar{x}}_{\left(F\right)}\right)}^{2}}{n_{2}-1}$$
where $s_{1}^{2}$ and $s_{2}^{2}$ are the variances of various joints for males and females.

From Table 3, all twenty joints are statistically significant at 1% level of significance, because *p*-value of all joints is zero and less than 1%—this suggests that there is clearly difference between males and females at each joint during a casual walk. With the help of the *t*-test, it is concluded that all joints of males and females play different roles during gait.

#### 3.3.3. Comparison of Gender Joints Based on ANOVA Technique

*t*-test is used for the comparison of two joints simultaneously. To compare more than two joints at the same time, the authors used analysis of variance (ANOVA) technique. Table 4 shows the result of ANOVA for particular joints during gait of both males and females—Twenty joints of males compared with twenty joints of females; k = 40 and d.f = 39. Further, = 1440 and k = 40, therefore, an error d.f = n − k = 1400, sum of square (SS) and mean square (MS), respectively. The *p*-value is less than 5% and difference between various joints is significant, i.e., there is significant difference between males and females at joint hip center. It suggests that, at joint hip center, the female contribution is higher than males, which further concludes that the contribution of the hip center during walking is different for both males and females, as shown in Table 3. The *t*-test and ANOVA technique provides us the same conclusion. The constant term k denotes the total numbers of joints, i.e., male = 20 and female = 20.

### 3.4. Feature Selection

We first calculate the adjusted mean and adjusted standard deviation to determine the Cronbach’s alpha value. By evaluating a feature’s consistency, or how close together a set of objects is, we can use Cronbach’s alpha to assess it.

The Cronbach’s alpha yields an average result of 99.74% as shown in Table 5. It indicates that the data can be analyzed very effectively.

### 3.5. Classification Using Machine Learning

Machine learning classification can often capture some key information to represent subjects’ identities, which other classifiers cannot capture, by iteratively feeding gait features and correlating labels into machine learning models in the training phase. Regression models are a well-known machine learning classifications for gait biometrics. The relationship between the categorical target variable and one or more independent factors is measured using logistic regression. When there are just two alternative outcomes for a goal variable, it can be helpful (in other words, it is binary). In order to predict the target variable classes, binary logistic regression classification uses one or more predictor variables that may be continuous or categorical. With the aid of this technique, it is possible to pinpoint crucial variables (Xi) that have an impact on the dependent variable (Y) as well as the nature of their interactions. Binary logistic regression model can be represented mathematically as
(22)$$P_{1}=\frac{e^{\left(a+bX\right)}}{1+e^{\left(a+bX\right)}}$$
(23)$$P_{0}=\frac{1}{1+e^{\left(a+bX\right)}}$$

The main advantage of the proposed system is to identify the gender (male/female) of the walking subject on the basis of gait 3D joints positions detected via MS Kinect controller. The proposed system calculates the values of $P_{1}$ and $P_{0}$ using Equations (22) and (23). When the value of $P_{1}$ exceeds the threshold value, the walking person will be a male, and when the value of $P_{0}$ exceeds the threshold value, the walking person will be a female. To compare the results of the fitted binary logistic regression model, we used the threshold value of the binary logistic regression model for gender classification.

## 4. Experimental Result

To evaluate our proposed method, we used a dataset of 373 different individuals wearing their typical shoes and clothes with different ages and gender. Our setup placed the Kinect camera on a tripod at a height of 1.0 m in a typical room measuring 6 m × 8 m with normal daylight. On a PC with a 64 GB RAM and an Intel Core i3 312 GHZ processor, the recorded data were processed. The training computations take the walking of subjects for only a few seconds to complete. The MS Kinect tracks the subjects’ 3D joints positions at a rate of 30 frame per seconds (fps). We recorded people walking toward and away from the camera in front of the sensor in order to capture lengthy sequences. As a result, each walking sequence had between 30 and 150 frames. Different step sizes and speeds among different participants caused variations in sequence duration. As for 3D joint detection, the visibility of the full body joints is mandatory in front of the MS Kinect; therefore, the maximum distance between the MS Kinect and walking subject is defined as 6 feet (see Figure 4).

The main factor to optimize the distance to be 6 feet was the “height of the person” and “resolution (640 × 480) of the camera”. Because when a tall person walks near to the camera, due to their height, some of the body joints are occluded, which affects the accuracy due to the detection of few joints. In contrast, a person walking beyond the working area of the MS Kinect can result in the false detection of 3D joints. The detection of the walking person has no effect on the distance between subject and MS Kinect, but just for the detection of full body joints. For a better depiction of the data, the data also include additional details, such as participants’ age and gender. Figure 5 and Figure 6 show screenshots of the gender detected in real time from their gait in different direction. For experimental analysis, 100 volunteer subjects (50 male and 50 female) participated in the experimental study.

### 4.1. Classification Accuracy

We have evaluated several parameters and features of our method in order to fully assess it. We assess the efficiency of joint combinations. The accuracy’s detected by our system, using different set of joints such as 9, 12, 14, 16, 18, and 20 were 93.0%, 95.0%, 90.0%, 96.0%, 97.0%, and 98.0%, respectively. The best accuracy was 98.0% when all 20 joints in our suggested system were used. We consider several combinations of walking sequences for training and testing in order to choose the optimal model for gender classification. For each set, we maintained a minimum difference of two to three joints. By contrasting all of the male and female joints, we find a natural overall difference. The complete set of joints greatly enhanced the output; however, the variance is minimal and the rate of increase is moderate, demonstrating the robustness of our classifier to the training data. To measure the accuracy of classification, we utilize an accuracy definition as
(24)$$accuracy=\frac{T_{P}+T_{N}}{T_{P}+F_{P}+T_{N}+F_{N}}$$

Our classification accuracy for each of the classes is summarized in Table 6. The proposed technique achieves the same accuracy for males as well as females, but comparatively it achieves better accuracy than other systems.

The covariates are the combined averages of all human joints. The walking person’s gender (y = 1, 0) is the dependent variable. Additionally, we evaluate that all of the fitted binary logistic regression model’s coefficients are equal to zero using the G-statistic of 71.463, the DF of 20, and the *p*-value of 0.000. Further, we determine the statistical significance of each coefficient parameter at the 5% level. The predicted values of the best-fit binary logistic regression model are categorized in confusion matrix (see Figure 7). The model successfully predicted 49 out of 50 males, and 49 females out of 50 females. Only one man and one woman had incorrect predictions. By comparing threshold values (0.5), the model predicts gender status 98.0% accurately.

Accuracy will decrease with larger standard errors, and will increase with lower standard errors. These joints provide better gender detection because the its standard errors are minimum. Moreover, the *p*-value for each joints is less than 0.05, which indicates that the joints are statistically significant in prediction of a gender as shown in Table 7. The standard error of joints can be computed using formula
(25)$$S.E\left({\bar{X}}_{j}\right)=\frac{\sigma_{j(x)}}{\sqrt{n}}$$

#### 4.1.1. Comparison of the Proposed System with Different Conventional Algorithms

We tested the result of the proposed system using a number of conventional algorithms, such as naive Bayes, multi-layer perception (MLP), random forest (RF), and k-nearest neighbor (KNN). The comparison results are shown in Table 8. Performance evaluation parameters are used to generate the experimental results of gender classification as well as the parameters for performance evaluation. Classification accuracy and true positive rate (recall) for male and female individuals have been considered as evaluation criteria. In the conventional algorithms, the best performance is shown by the KNN algorithm, with a classification accuracy of 92.0%. In comparison to the conventional algorithms, the classification accuracy of the proposed system is 98.0%, which is better than all the existing systems. In the conventional methods, most of the systems showed unbalanced recall for males and females, which indicates that the data are more difficult to classify. On the other hand, the proposed system is better because it shows a balanced recall for both males and females.

To the best of the author’s knowledge, very limited datasets are publicly available for gait-based gender recognition captured by Microsoft Kinect. After an extensive literature review, we found one public dataset (UPCVgait) of gait-based data using Microsoft Kinect for gender recognition. We applied the proposed model on the above dataset for gender classification and compared it with the other conventional methods [55]. As shown in Table 9, the existing system produced an accuracy of 96.67% [55]. The proposed system outperformed the existing system with an accuracy of 98.5% by using the same dataset (see Table 8).

#### 4.1.2. Discussion

The statistical findings of this research reveal that coefficient of variations for both male and female joints during gait shows significant differences. The coefficient of variation at joint head for males is 10.31 (see Table 1), which shows less variability as compared to others joints of male. Therefore, the joint *head* of males is more consistent during walking. On the contrary, the coefficient of variation at the joint *head* of female is 9.05, which also shows less variability compared to the remaining joints of females. So, the joint *head* is also more consistent for females during casual walk. Similarly, the coefficient of variation at joint *foot-left* of males is 17.58, which shows high variability compared to the remaining joints of males. Therefore, the male performance at joint *foot-left* varies from person to person. For *foot-left* of females, the coefficient of variation is 17.01, which shows less variability as compared to joint foot left of males. Therefore, the female performance at joint *foot-left* is consistent compared to males. The coefficient of variations at joint *foot-right* for males and females are 16.34 and 17.82, respectively, which shows that male performance is more consistent than female at joint *foot-right*.

To compare all joints using *t*-test, there is a significant difference between male and female joints. *p*-values for all joints are zero (see Table 3) which indicates that male and female joints play different roles during gait.

According to analysis of correlation, both males and females widely move in the *X* and *Z* coordinates, as the correlation is high and highly significant at *X* and *Z* coordinates, i.e., for both males and females at *X* and *Z*-coordinates because the *p*-values for each joints is zero. On the contrary, the correlation is weak and less than 50% for most joints of males and females at *X* and *Y*-coordinates as well as *Y* and *Z*-coordinates. The correlation of joints elbow-left and hip-left for male and head of female are more than 90%, which means that these joints are highly significant (see Table 2). According to analysis of correlation there is quit difference between male and female joints. Outcomes of *t*-test show that all joints of male and female are significantly different from each other. ANOVA technique was applied and provided the same conclusion for reduction in type-I error and for comparison of all joints of male and female. It showed that all joints of males and females are statistically significant at 1% and 5% level of significance.

Analysis and results of *t*-test, ANOVA, and correlation suggest that male and female joints have different contributions during walking. Further more, the Cronbach’s alpha yields a result of 99.75%, which suggests the reliability of data. Finally, the gender of the walking subjects are detected with an accuracy of 98.0% using a machine learning approach.

## 5. Conclusions and Future Work

In this paper, a machine learning approach for gait-based gender classification using all body joints is proposed. The proposed method included several stages, such as gait feature extraction based on three dimensional (3D) features, feature selection, and human gender classification. The proposed system extracted the 3D joints positions of all human joints using Kinect sensor, which were compared for the identification of gender during walk. Different statistical tools such as Cronbach’s alpha, correlation, *t*-test, and ANOVA techniques were exploited to select significant joints. The Cronbach’s alpha technique yielded an average result of 99.74%, which indicates the reliability of joints. Similarly, the correlation results indicated that there is significant difference between male and female joints during gait. *t*-test and ANOVA approaches demonstrated that all twenty joints are statistically significant for gender classification, because the *p*-value for each joint was zero and less than 1%. It was observed that all male joints contributed differently than female joints during gait. The results showed that the *head* is consistent for both male and female during walk. Furthermore, the results proved that the joints *foot-left* and *foot-right* of males and females vary more from individual to individual. The joints *hand-left*, *knee-right*, *hip-center*, and *spine* of all males played the same role during gait. Further, the joint *spine*, *shoulder-center*, *head*, and *shoulder-right* played the same contribution for all female during gait. According to the coefficient of variation of male and female joints, male joints are more consistent than female joints during walking, which showed that there is a large variance in female joints as compared to male joints. There was high variation in males at joint knee left during walking, which suggested that contribution of joint *knee-left* of males varies from person to person. There was high variation in joints *foot-right*, *ankle-right*, and *foot-left* of females, which indicates that female contribution varies from female to female. Finally, the gender of the walking person was correctly identified with an accuracy of 98.0% using all body joints. To demonstrate the improved performance of the proposed system, experimental results are compared with deep learning as well as with conventional machine learning techniques. The comparative studies demonstrated the robustness of the proposed machine learning approach, which outperforms the state-of-the-art models. The majority of research on gait-based gender recognition that have been published in the literature have relied on very small sample sizes, which may have an impact on gender identification.

In future work, the researchers would strive further to strengthen the results by studding a much larger database. The proposed method will investigate which joints of males and females are more significant in running, jumping, and sports. Furthermore, these results can be used for modeling human identification.

## Figures and Tables

**Figure 1 sensors-22-09113-f001:**
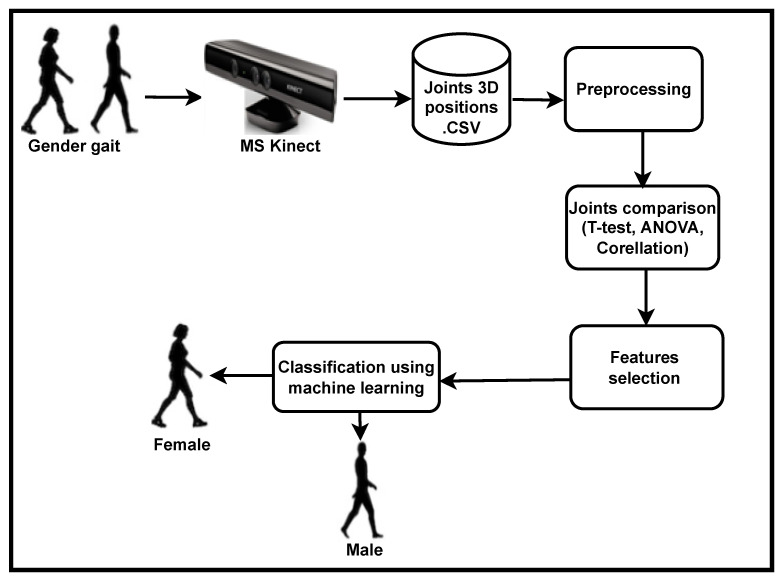
Detail system model of the proposed system.

**Figure 2 sensors-22-09113-f002:**
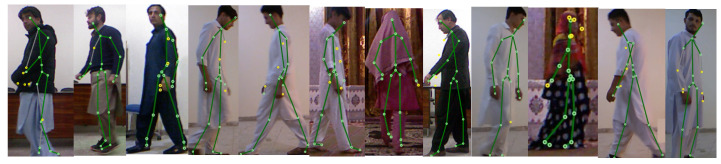
Walking of subjects in different directions.

**Figure 3 sensors-22-09113-f003:**
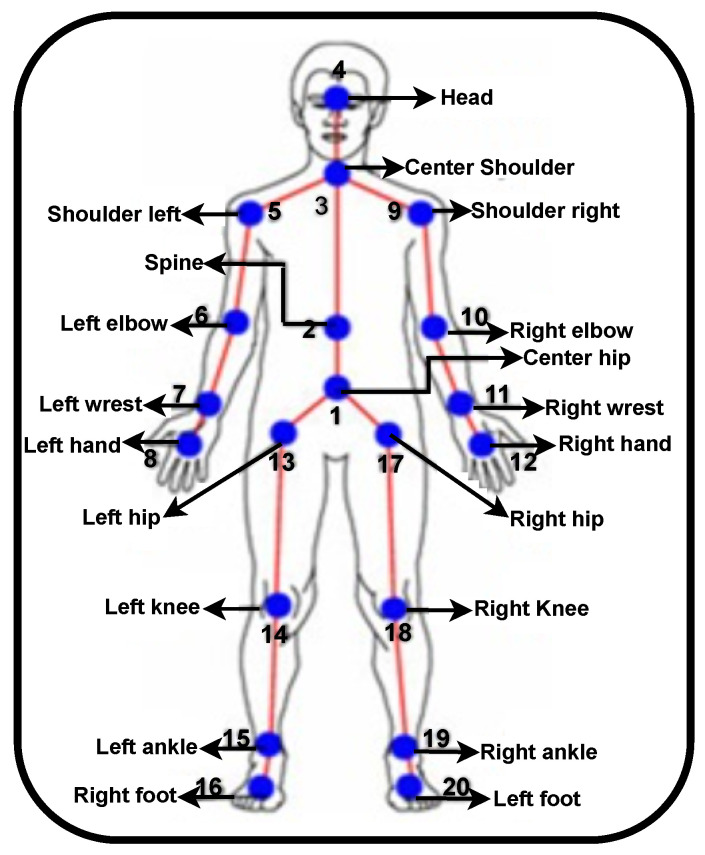
Various human joints.

**Figure 4 sensors-22-09113-f004:**
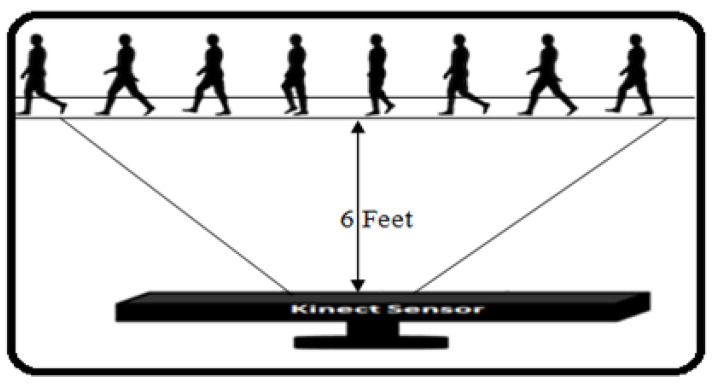
Experimental setup using MS Kinect.

**Figure 5 sensors-22-09113-f005:**
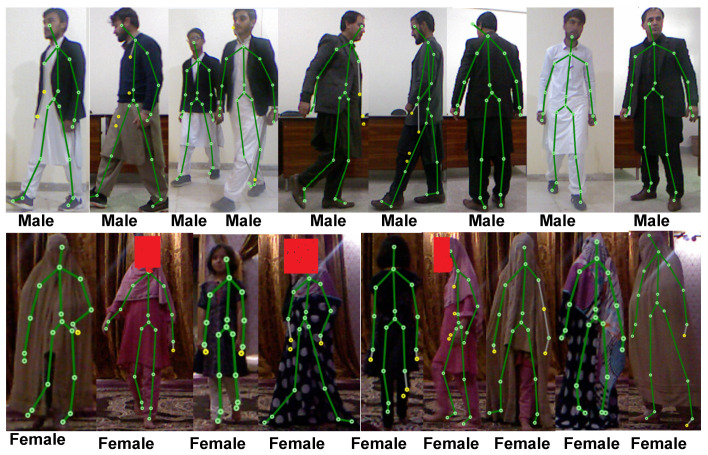
Walking of different subjects from different directions.

**Figure 6 sensors-22-09113-f006:**
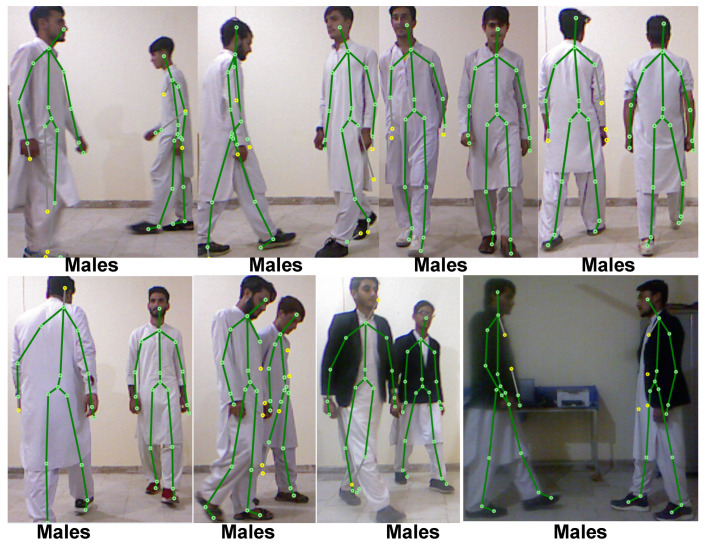
Walking of more than one subjects at the same time from different directions.

**Figure 7 sensors-22-09113-f007:**
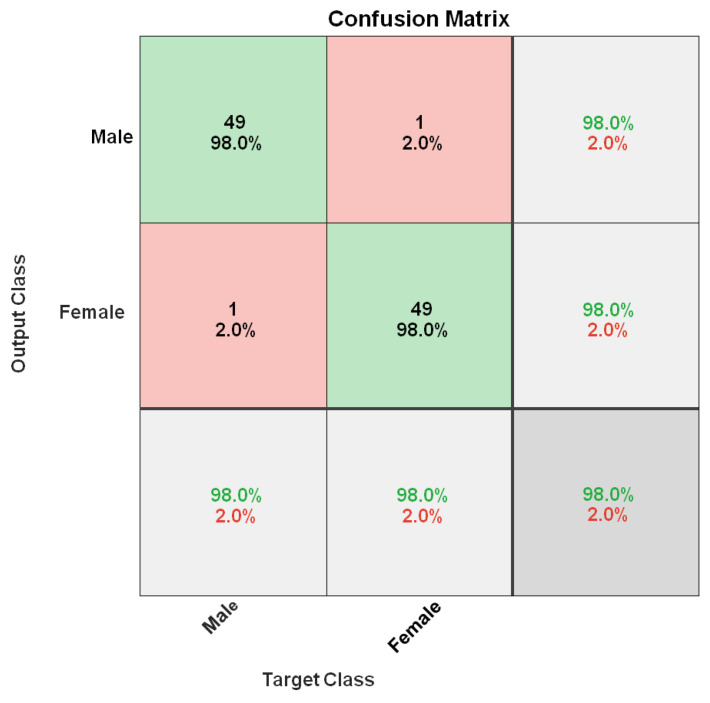
Confusion matrix.

**Table 1 sensors-22-09113-t001:** The mean, standard deviation (S.D), and coefficient of variation (C.V) for various joints of male and female.

	Male			Female		
**Joints**	**Mean**	**S.D**	**C.V**	**Mean**	**S.D**	**C.V**
Hip Center	0.9918	0.1154	11.63	1.1906	0.1489	12.50
SPINE	1.0233	0.1154	11.63	1.1906	0.1489	12.50
Shoulder Center	1.1693	0.1232	10.53	1.2237	0.1439	11.76
HEAD	1.2023	0.1240	10.31	1.3833	0.1252	9.05
Shoulder Left	1.1382	0.1272	11.18	1.2883	0.1515	11.76
Elbow Left	1.0504	0.1266	12.06	1.2220	0.1785	14.61
Wrest Left	0.9630	0.1264	13.12	1.1724	0.1841	15.70
Hand Left	0.9343	0.1278	13.67	1.1553	0.1894	16.39
Shoulder Right	1.1114	0.1204	10.83	1.2981	0.1301	10.02
Elbow Right	1.0043	0.1190	11.85	1.2084	0.1448	11.98
Wrest Right	0.9049	0.1180	13.04	1.1245	0.1558	13.85
Hand Right	0.8770	0.1176	13.41	1.1011	0.1576	14.32
HIP Left	0.9738	0.1172	12.03	1.1673	0.1529	13.10
Knee Left	0.8130	0.1415	17.40	1.0458	0.1602	15.31
Ankle Left	0.7400	0.1224	16.53	0.9787	0.1615	16.50
Foot Left	0.7054	0.1240	17.58	0.9499	0.1616	17.01
HIP Right	0.9724	0.1155	11.88	1.1754	0.1481	12.60
Knee Right	0.8486	0.1160	13.67	1.0804	0.1576	14.59
Ankle Right	0.7607	0.1172	15.40	1.0120	0.1733	17.13
Foot Right	0.7217	0.1180	16.34	0.9827	0.1751	17.82

**Table 2 sensors-22-09113-t002:** The Pearson correlation between various joints in 3D coordinates system (*x*, *y*, and *z*).

	Male			Female		
**Joints**	* **X** * **and** * **Y** *	* **Y** * **and** * **Z** *	* **X** * **and** * **Z** *	* **X** * **and** * **Y** *	* **Y** * **and** * **Z** *	* **X** * **and** * **Z** *
HIP-C	0.84 (0.00)	0.88 (0.00)	0.90 (0.00)	0.14 (0.00)	0.20 (0.00)	0.80 (0.00)
HIP-L	0.81 (0.00)	0.88 (0.00)	0.94 (0.00)	0.01 (0.79)	0.01 (0.68)	0.84 (0.00)
HIP-R	0.82 (0.00)	0.76 (0.00)	0.93 (0.00)	0.24 (0.00)	0.19 (0.00)	0.60 (0.00)
Spine	0.86 (0.00)	0.84 (0.00)	0.90 (0.00)	0.22 (0.00)	0.16 (0.00)	0.85 (0.00)
Shoulder-C	0.65 (0.00)	0.52 (0.00)	0.92 (0.00)	0.29 (0.00)	0.16 (0.00)	0.88 (0.00)
Shoulder-L	0.60 (0.00)	0.49 (0.00)	0.95 (0.00)	0.04 (0.27)	−0.04 (0.24)	0.86 (0.00)
Shoulder-R	0.57 (0.00)	0.16 (0.00)	0.92 (0.00)	0.02 (0.61)	0.31 (0.00)	0.70 (0.00)
Head	0.70 (0.00)	0.66 (0.00)	0.90 (0.00)	0.32 (0.00)	0.27 (0.00)	0.90 (0.00)
Elbow-L	0.40 (0.00)	0.31 (0.00)	0.95 (0.00)	0.18 (0.00)	0.04 (0.26)	0.87 (0.00)
Elbow-R	0.39 (0.00)	0.54 (0.00)	0.93 (0.00)	−0.47 (0.00)	−0.25 (0.00)	0.82 (0.00)
Wrest-L	0.16 (0.00)	0.25 (0.00)	0.94 (0.00)	0.16 (0.00)	0.52 (0.00)	0.79 (0.00)
Wrest-R	0.56 (0.00)	0.45 (0.00)	0.96 (0.00)	−0.75 (0.00)	−0.22 (0.00)	0.77 (0.00)
Hand-L	0.08 (0.03)	0.15 (0.00)	0.93 (0.00)	0.08 (0.03)	0.49 (0.00)	0.72 (0.00)
Hand-R	0.49 (0.00)	0.32 (0.00)	0.91 (0.00)	−0.72 (0.00)	−0.29 (0.00)	0.80 (0.00)
Knee-L	0.35 (0.00)	0.49 (0.00)	0.90 (0.00)	−0.15 (0.00)	−0.37 (0.00)	0.90 (0.00)
Knee-R	0.41 (0.00)	0.44 (0.00)	0.91 (0.00)	−0.06 (0.09)	−0.13 (0.00)	0.79 (0.00)
Ankle-L	−0.49 (0.00)	−0.27 (0.00)	0.94 (0.00)	−0.44 (0.00)	−0.45 (0.00)	0.87 (0.00)
Ankle-R	−0.49 (0.00)	0.03 (0.49)	0.93 (0.00)	−0.47 (0.00)	−0.25 (0.00)	0.82 (0.00)
Foot-L	−0.59 (0.00)	−0.46 (0.00)	0.94 (0.00)	−0.54 (0.00)	−0.51 (0.00)	0.88 (0.00)
Foot-R	−0.53 (0.00)	0.03 (0.44)	0.93 (0.00)	−0.39 (0.00)	−0.32 (0.00)	0.88 (0.00)

**Table 3 sensors-22-09113-t003:** The results of the two-sample *t*-test used for comparison between male and female joints during walk.

Joints	DF	Pooled S.D	T-Value	*p*-Value
Hip Center	70	0.1222	6.61	0.000
SPINE	70	0.1214	−6.71	0.000
Shoulder Center	70	0.1222	−5.83	0.000
HEAD	70	0.1225	−6.12	0.000
Shoulder Left	70	0.1320	4.81	0.000
Elbow Left	70	0.1391	−5.10	0.000
Wrest Left	70	0.1410	6.03	0.000
Hand Left	70	0.1437	−6.21	0.000
Shoulder Right	70	0.1210	6.40	0.000
Elbow Right	70	0.1238	−6.81	0.000
Wrest Right	70	0.1261	−7.12	0.000
Hand Right	70	0.1262	−7.21	0.000
HIP Left	70	0.1246	−6.30	0.000
Knee Left	70	0.1432	6.63	0.000
Ankle Left	70	0.1304	−7.38	0.000
Foot Left	70	0.1315	7.50	0.000
HIP Right	70	0.1220	−6.75	0.000
Knee Right	70	0.1252	−7.53	0.000
Ankle Right	70	0.1307	−7.82	0.000
Foot Right	70	0.1317	−8.03	0.000

**Table 4 sensors-22-09113-t004:** The result of analysis of variance (ANOVA).

Source of Variation	DF	SS	MS	F-Value	*p*-Value
Joints	39	42.1105	1.0798	64.66	0.000
Error	1400	23.3802	0.0167	—	—
Total	1439	65.4906	—	—	—

**Table 5 sensors-22-09113-t005:** The results of adjusted mean (Adj.mean), adjusted standard deviation (Adj.SD), and Cronbach’s alpha for reliability of human body joints.

Joints	Adj.Mean	Adj.SD	Cronbach’s Value
${\bar{X}}_{C}\left(Hip_{C}\right)$	18.941	2.986	0.9973
${\bar{X}}_{C}\left(Spine\right)$	18.909	2.986	0.9973
${\bar{X}}_{C}\left(Sh_{C}\right)$	18.801	3.021	0.9979
${\bar{X}}_{C}\left(Head\right)$	18.735	2.994	0.9974
${\bar{X}}_{C}\left(Sh_{L}\right)$	18.807	2.993	0.9974
${\bar{X}}_{C}\left(E_{L}\right)$	18.889	2.982	0.9974
${\bar{X}}_{C}\left(W_{L}\right)$	18.967	2.972	0.9974
${\bar{X}}_{C}\left(H_{L}\right)$	18.992	2.968	0.9974
${\bar{X}}_{C}\left(Sh_{R}\right)$	18.824	2.992	0.9973
${\bar{X}}_{C}\left(E_{R}\right)$	18.927	2.984	0.9973
${\bar{X}}_{C}\left(W_{R}\right)$	19.022	2.979	0.9973
${\bar{X}}_{C}\left(H_{R}\right)$	19.049	2.977	0.9973
${\bar{X}}_{C}\left(Hip_{L}\right)$	18.960	2.985	0.9973
${\bar{X}}_{C}\left(K_{L}\right)$	19.111	2.968	0.9976
${\bar{X}}_{C}\left(A_{L}\right)$	19.182	2.971	0.9973
${\bar{X}}_{C}\left(F_{L}\right)$	19.215	2.969	0.9973
${\bar{X}}_{C}\left(Hip_{R}\right)$	18.959	2.985	0.9972
${\bar{X}}_{C}\left(K_{R}\right)$	19.074	2.976	0.9973
${\bar{X}}_{C}\left(A_{R}\right)$	19.158	2.967	0.9974
${\bar{X}}_{C}\left(F_{R}\right)$	19.194	2.964	0.9974

**Table 6 sensors-22-09113-t006:** Classification accuracy of the proposed system with different set of joints.

Gender	09 Joints	12 Joints	14 Joints	16 Joints	18 Joints	20 Joints
Male	92.0	94.0	90.0	96.0	98.0	98.0
Female	94.0	96.0	90.0	96.0	96.0	98.0
Both	93.0	95.0	90.0	96.0	97.0	98.0

**Table 7 sensors-22-09113-t007:** Standard error, coefficient, and *p*-Value of binary logistic regression model.

Predictors (Joints)	Coefficients	Standard Error	*p*-Value
Constant	18.30	23.6511	0.013
${\bar{X}}_{C}\left(Hip_{C}\right)$	20.2	42.1786	0.043
${\bar{X}}_{C}\left(Spine\right)$	55.5	36.1421	0.021
${\bar{X}}_{C}\left(Sh_{C}\right)$	45.3	47.0012	0.032
${\bar{X}}_{C}\left(Head\right)$	10.75	55.5609	0.040
${\bar{X}}_{C}\left(Sh_{L}\right)$	−12.0	30.9800	0.026
${\bar{X}}_{C}\left(E_{L}\right)$	22.7	19.3451	0.029
${\bar{X}}_{C}\left(W_{L}\right)$	33.3	29.6555	0.032
${\bar{X}}_{C}\left(H_{L}\right)$	28.9	42.7764	0.035
${\bar{X}}_{C}\left(Sh_{R}\right)$	36.0	23.2213	0.026
${\bar{X}}_{C}\left(E_{R}\right)$	29.1	49.3244	0.029
${\bar{X}}_{C}\left(W_{R}\right)$	50.4	32.9212	0.034
${\bar{X}}_{C}\left(H_{R}\right)$	44.6	40.6665	0.038
${\bar{X}}_{C}\left(Hip_{L}\right)$	−11.5	60.6542	0.020
${\bar{X}}_{C}\left(K_{L}\right)$	20.8	55.8721	0.033
${\bar{X}}_{C}\left(A_{L}\right)$	26.9	44.8111	0.036
${\bar{X}}_{C}\left(F_{L}\right)$	45.2	34.5444	0.038
${\bar{X}}_{C}\left(Hip_{R}\right)$	52.3	33.2111	0.040
${\bar{X}}_{C}\left(K_{R}\right)$	37.8	35.5433	0.041
${\bar{X}}_{C}\left(A_{R}\right)$	39.1	45.4456	0.046
${\bar{X}}_{C}\left(F_{R}\right)$	10.6	22.5467	0.045

**Table 8 sensors-22-09113-t008:** Comparison of the proposed system with the conventional systems based on classification accuracy.

Algorithm	Recall Male (%)	Recall Female (%)	Accuracy (%)
Naive Bayes	80.0	70.0	75.0%
MLP	84.0	70.0	77.0%
RF	92.0	88.0	90.0%
KNN	94.0	90.0	92.0%
Our	98.0	98.0	98.0%

**Table 9 sensors-22-09113-t009:** Comparison of classification accuracy between the proposed system and the conventional system.

Classifier	Dataset	Accuracy (%)
SVM [55]	UPCVgait	96.67%
Proposed Method	UPCVgait	98.5%

## Data Availability

Not applicable.

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
