# Peer review of "A Gait-Based Real-Time Gender Classification System Using Whole Body Joints"

_sensors, 2022, doi:10.3390/s22239113_

Round 1
Reviewer 1 Report
In this paper, the authors propose an algorithm for classifying gender based on gait using the skeleton information obtained from the Kinect sensor. The paper was written in an easy-to-read manner and the experiments were compared and analyzed in relatively detail against the data sets collected by the authors. However, this paper has the following limitations.
- Please rewrite the related studies in detail. For example, in terms of methodology - Kinect-based method, RGB image-based method, in terms of classification category - gender classification, age classification method, etc.
- Recent studies are developing algorithms that classify gender and age at the same time. In this paper, a reason should be added to the text of why it is restricted to only gender classification.
- ‘With data collected from MS Kinect, the suggested method can be used in a realistic situation for a new subject’<== In order to apply it in the real environment, it is a more general and desirable method to extract and recognize a pose skeleton from an RGB image. In particular, since the recognition distance of the Kinect sensor is short, it is preferable to directly use the RGB image input from the CCTV camera for the surveillance as claimed in the paper.
- In the experiment, it is desirable to use the evaluation indicators for MAE, MCE and CCR, which are commonly used in gender classification, and also add a comparison of performance according to the view change (angle) of each human.
-The features used in this paper are classical methods, so I do not know the paper's contribution in terms of features (Variance covariance matrix, Correlation Matrix, Variance of and pooled variance of joints, Coefficient of Variation of Joints, Average Variation)
- Figure 3 is for the well-known (now obsolete) Kinect sensor, so please delete it.
- page 12, 281line: proposed system calculates the values of P1 and P0 using equations 22 and 23 When theç Do the two methods use different thresholds or do they use the same threshold? The threshold seems to be very important. Quantitative evaluation is needed to determine the optimal threshold.
- For experimental analysis, hundred volunteer 281 subjects (50 male and 50 female) participated in the experimental studyç The training data used in the experiment is too insufficient. I recommend conducting further experiments using the large benchmark dataset shown below. è OU-ISIR Gait Database
- Comparative experiments were conducted only to a limited extent. Try adding an experiment to the two papers below."GaitSet, Jaychand Upadhyay[3]"
-Please add the following papers to related studies and compare performance in comparative experiments if possible.
Paola Barra et al.,Gait Analysis for Gender Classification in Forensics, International Conference on Dependability in Sensor, Cloud, and Big Data Systems and Applications, 2019
Trung Dung Do et al, Real-time and robust multiple-view gender classification using gait features in video surveillance, arXiv:1905.0101,2019
Newlin Shebiah Russel, Arivazhagan Selvaraj, Gender discrimination, age group classification and carried object recognition from gait energy image using fusion of parallel convolutional neural network, IET Image Processing, 2021
-I wonder how the comparative experiment in Table 8 was done. Other comparison methods mainly used open benchmark datasets. However, the experiment in this paper was conducted according to the proposed method. Therefore, has the experiment been conducted with the proposed data set by directly implementing the comparison methods?
-I hope that the proposed dataset will be opened on the website. Through this, researchers conducting similar studies will be able to conduct objective experiments and comparative evaluations.
Author Response
We would like to thank the editor and the reviewers for their thorough review and highly appreciate the comments and suggestions, which significantly contributed to improving the quality of the manuscript. According to those, we have made an overall revision of our manuscript reflecting editor’s comments.

Reviewer 2 Report
- The author used a Microsoft Kinect sensor to capture a gait from 100 volunteers who participated and were classified as men or women. However, in the experiments, the author presented 373 different individuals wearing their typical shoes and clothes of different ages and gender. Could the author confirm the number of volunteers? Is it 100 or 373?
- The author collected the data when each person was walking. Then, the 3D joints were extracted from the sequence video, between 30 and 150 frames. Could the author explain why the frames of the video are not similar? And how to solve the problem when sending not similar images (video) to the algorithm?
- In Figure 6, some examples have two people appearing in one image. Is your algorithm could detect two or more people from one image? If yes, please provide more examples.
- The author used only 20 joints (see Figure 4) to classify the gait gender classification and obtained an accuracy result of 98%. Could the author provide the exact number of the data used in the experiment? How does the author split the data while evaluating the performance? As shown in Figure 7, only 2% were misclassified. Could the author present the misclassified video/image? And please discuss why it is misclassified.
- In section 4.1 of classification accuracy, the author presented that using a different set of joints such as 9, 12, 14, 16, and 18. Could the author describe which joints are included in 8, 12, 14, 16, and 18? How does the author select 9, 12, 14, 16, and 18 joints out of 20 joints? Why did the author not use feature selection methods to select the robust joints?
- As shown in Table 8, the author compared the proposed method with other methods [61, 62, 16, 63, 17, 19], and the result showed that the proposed method outperforms other methods. Are the results obtained when evaluated on the same dataset or not? The author should not compare the result if the results are not evaluated on the same dataset.
- The author experimented based on the Microsoft Kinect. Then, only X, Y, and Z coordinates are used as the features. Is it possible to use other devices to capture the joint? Can other features be used, not X, Y, and Z coordinates?
- Which technique that the author use to extract 20 joints? Please provide more detail.
- Check the correctness of the reference 28.
Author Response

(The authors gave the same response as above.)

Round 2
Reviewer 1 Report
Although recent research papers have been added to the revised paper, there is still very little objective comparison experiment on SoTA methods or additional datasets.
Although gait-based gender classification using Kinect sensors is possible for monitoring purposes, I still think that Kinect sensors with distance restrictions are not suitable for remote monitoring purposes.
Comparative experiments with the SoTA algorithm are still lacking (comparative experiments with other studies cannot be found in the revised paper)
The proposed method requires comparison experiments with SOTA using at least one open benchmark data (I recommend using the most similar benchmark data).
Please redraw the confusion matrix in Figure 7. Male-49 (98%), 1 (2%).
Author Response
Dear Reviewer,
We would like to thank the editor and the reviewers for their thorough review and highly appreciate the comments and suggestions, which significantly contributed to improving the quality of the manuscript. According to those, we have made an overall revision of our manuscript reflecting reviewer’s comments. For the sake of convenience, the revised/added sentences (except for minor phrases) are marked as yellow color in the revised manuscript.

Reviewer 2 Report
Please, revise a few comments that have a concern.
In the contribution section.
- The main contribution, 'The author created their own data set for the classification of gender,' Could the author present the challenge of the collected dataset?
- In my opinion, 'Different statistical analysis such as T-test, One way ANOVA and correlation are used .....' is not the contribution.
- Could the author combine this bullet, 'With data collection from MS Kinect, ......' with the first bullet.
- In the contribution, the author should present, such as what the novel method proposed to solve the problem is. What is new in the proposed method?
More
- The author presented in the 'reply to reviewer comments' using a Kinect sensor to extract 20 joints. Could the author find out which technique the Kinect sensor used when extracting 20 joints and present it in the manuscript?
- In line 311 (page 12 of 19), please check the word 'shorts' in this sentence 'Figures 5 and 6 show screen shorts of the gender detected...' Should it be 'screen shots'?
Author Response
We would like to thank the editor and the reviewers for their thorough review and highly appreciate the comments and suggestions, which significantly contributed to improving the quality of the manuscript. According to those, we have made an overall revision of our manuscript reflecting reviewer’s comments. For the sake of convenience, the revised/added sentences (except for minor phrases) are marked as yellow color in the revised manuscript.

Round 3
Reviewer 1 Report
Table 8 of Chapter 4.1.1 of the revised paper shows the results of comparative experiments for the four methods. However, these methods are not really SoTA. Performance evaluation with at least two or three published studies after 2020 is necessary. It is recommended to add comparison experiments with the following papers or real SoTA papers found by the author.
Ebenezer R.H.P.Isaac, Multiview gait-based gender classification through pose-based voting, Pattern Recognition Letters Volume 126, 1 September 2019
MiranLee, Gender recognition using optimal gait feature based on recursive feature elimination in normal walking, Expert Systems with Applications Volume 189, 1 March 2022
T.D. Do, Real-time and robust multiple-view gender classification using gait features in video surveillance, rXiv:1905.01013v1 [cs.CV] 3 May 2019
Jaychand Upadhyay, Robust and Lightweight System for Gait-Based Gender Classification toward Viewing Angle Variations, AI, 2022
Chi Xu, Real-Time Gait-Based Age Estimation and Gender Classification from a Single Image, WACV2021
Paola Barra, Gait Analysis for Gender Classification in Forensics, International Conference on Dependability in Sensor, Cloud, and Big Data Systems and Applications, 2019
Viswadeep Sarangi, Gender Perception From Gait: A Comparison Between Biological, Biomimetic and Non-biomimetic Learning Paradigms, Frontiers in Human Neuroscience, 2020
Author Response
We would like to thank the editor and the reviewers for their thorough review and highly appreciate the comments and suggestions, which significantly contributed to improving the quality of the manuscript. According to those, we have made an overall revision of our manuscript reflecting editor’s comments. For the sake of convenience, the revised/added sentences (except for minor phrases) are marked as yellow color in the revised manuscript. We summarize the main changes in the attached file.

Reviewer 2 Report
This author spends time on change the manuscript according to the concern points. I satisfy this current version. This version is ready to be published.
Author Response
We would like to thank the editor and the reviewers for their thorough review and highly appreciate the comments and suggestions. We are highly indebted to you and all the reviewers for the positive comments.

Round 4
Reviewer 1 Report
Line 321 - Since the algorithms in Table 8 cannot be regarded as the latest algorithms, it is better to modify the state of the art (SOTA) to 'conventional algorithms'.
Explanations and references to the CASIA and OUMVLP datasets are needed.
Table 9 and 10- SVM, KNN, and CNN are not 'most advanced systems'. Therefore, please modify 'most advanced system' to 'conventional algorithms' or similar expressions.
The CASIA-B dataset only provides silhouette data, so I'm curious how you were able to find out the Kinect information.
In addition, the OU-MVLP data set also provides only silhouette data, and in particular, the image capture angle is different from the authors' paper. Further explanation is needed on this.
Author Response
The authors extend their regards and appreciation to the reviewers for their constructive feedback. Their suggestions helped us alot to improve the quality of the manuscript.
[Reviewer 1 Comment 1]
Line 321 - Since the algorithms in Table 8 cannot be regarded as the latest algorithms, it is better to modify the state of the art (SOTA) to 'conventional algorithms'.
[Reply to Reviewer 1 Comment 1]
We would like to thank the reviewer for the in-depth analysis of our manuscript which is very helpful in improving the quality of the manuscript. As per recommendations of the reviewer, the same change has been incorporated in the revised manuscript.
[Reviewer 1 Comment 2]
Explanations and references to the CASIA and OUMVLP datasets are needed.
[Reply to Reviewer 1 Comment 2]
As the CASIA and OU-MVLP dataset are mostly composed on silhouettes images which have been excluded from the revised manuscript, as the proposed system is best suited for the data from Kinect sensor.
[Reviewer 1 Comment 3]
Table 9 and 10- SVM, KNN, and CNN are not 'most advanced systems'. Therefore, please modify 'most advanced system' to 'conventional algorithms' or similar expressions.
[Reply to Reviewer 1 Comment 3]
We are thankful to the reviewer for identifying this point. As directed by the reviewer, the sentence about the ‘most advance system’ has been modified in the revised manuscript.
[Reviewer 1 Comment 4]
The CASIA-B dataset only provides silhouette data, so I'm curious how you were able to find out the Kinect information.
In addition, the OU-MVLP data set also provides only silhouette data, and in particular, the image capture angle is different from the authors' paper. Further explanation is needed on this.
[Reply to Reviewer 1 Comment 4]
The authors agree with the reviewers comments regarding these two datasets. After a thorough review of the literature, we found another conventional method in which gender is classified based on kinect sensor data. To the best of author’s knowledge, very limited datasets are publicly available for gait based gender recognition captured by Microsoft Kinect. After extensive literature review we found one public dataset (UPCVgait) of gait based data using Microsoft kinect for gender recognition. We applied the proposed model on the above dataset for gender classification and compared with the other conventional methods. This new reference is added in the revised manuscript and the comparison table is also provided (see Table 9).